# Debiased Machine Learning without Sample-Splitting for Stable Estimators

**Qizhao Chen**
Harvard University
Cambridge, MA 02138
qizhaochen@g.harvard.edu

**Vasilis Syrgkanis**
Stanford University
Stanford, CA 94305
vsyrgk@stanford.edu

**Morgane Austern**
Harvard University
Cambridge, MA 02138
morgane.austern@gmail.com

## Abstract

Estimation and inference on causal parameters is typically reduced to a generalized method of moments problem, which involves auxiliary functions that correspond to solutions to a regression or classification problem. Recent line of work on debiased machine learning shows how one can use generic machine learning estimators for these auxiliary problems, while maintaining asymptotic normality and root-$n$ consistency of the target parameter of interest, while only requiring mean-squared-error guarantees from the auxiliary estimation algorithms. The literature typically requires that these auxiliary problems are fitted on a separate sample or in a cross-fitting manner. We show that when these auxiliary estimation algorithms satisfy natural leave-one-out stability properties, then sample splitting is not required. This allows for sample re-use, which can be beneficial in moderately sized sample regimes. For instance, we show that the stability properties that we propose are satisfied for ensemble bagged estimators, built via sub-sampling without replacement, a popular technique in machine learning practice.

## 1 Introduction

A large variety of problems in causal inference and more generally semi-parametric inference can be framed as finding a solution to a moment condition:

$$M(\theta, g) \triangleq \mathbb{E}_Z[m(Z; \theta, g)] \qquad\qquad M(\theta_0, g_0) = 0$$

where $Z \in \mathcal{Z}$ is a vector of random variables that, apart from the target parameter $\theta_0 \in \Theta \subset \mathbb{R}^p$ of interest (we assume $p = O(1)$), also depends on unknown nuisance functions $g_0 \in \mathcal{G}$, which need to be estimated in a flexible manner from the data. This framework has a long history in the literature on semi-parametric inference [38, 26, 27, 47, 36, 51, 57, 12, 40, 49, 58, 13, 41, 1, 42, 2, 53, 37, 3], which analyzes the following two-stage estimation process with sample re-use, when having access to $n$ iid samples $\{Z_1, \ldots, Z_n\}$ and as $n$ grows, treating the target parameter dimension $p$ as a constant:

1. Obtain an estimate $\hat{g} \in \mathcal{G}$ of the nuisance function $g_0$ based on all the samples.

2. Return any estimate $\hat{\theta} \in \Theta$ that satisfies:

$$M_n(\hat{\theta}, \hat{g}) = o_p\left(n^{-1/2}\right) \qquad \text{with} \qquad M_n(\theta, g) \triangleq \frac{1}{n} \sum_{i=1}^{n} m(Z_i; \theta, g).$$

Given that the estimation of function $\hat{g}$ is a complex non-parametric problem, it will typically only satisfy slower than parametric error rates. Moreover, in high dimensional settings, when machine learning techniques are used to estimate $\hat{g}$, then the regularization bias of $\hat{g}$ will propagate to the final estimate $\hat{\theta}$, leading to non-regular estimates and the inability to construct confidence intervals.

36th Conference on Neural Information Processing Systems (NeurIPS 2022).

The literature on efficient semi-parametric estimation provides conditions on the moment function so that the influence of the estimation error of the nuisance function $\hat{g}$ is of second order importance and does not alter the distributional properties of the second stage [26, 13, 59, 7, 8, 9, 10, 31, 32, 33, 54, 46, 19, 43, 48, 28, 29, 30, 16, 61, 62]. This approach dates back to the classical work on doubly robust estimation and targeted maximum likelihood [50, 49, 56, 55, 39, 52] as well as the more recent work on locally robust or Neyman orthogonal conditions on the moment function [44, 45, 18, 6, 17], typically referred to as double or debiased machine learning. At a high-level, the earlier literature on semi-parametric inference shows that if the moment function satisfies some form of robustness to nuisance perturbations and, importantly, as long as the space $\mathcal{G}$ used in the first stage estimation is a relatively simple function class, in terms of statistical complexity, typically referred to as a Donsker function class, then the second stage estimate is root-$n$ consistent and asymptotically normal. Hence, one can easily construct confidence intervals for the target parameter of interest.

Crucially the recent literature on debiased machine learning alters the standard two-stage estimation algorithm to introduce the idea of sample-splitting, [11, 60]. In particular, instead of estimating $\hat{g}$ on all the samples, the recent work on debiased machine learning [17, 18, 22] estimates $\hat{g}$ on a separate sample, or for better sample efficiency invokes "cross-fitting," where we train a nuisance model on half the data and evaluate it in the second stage on the other half and vice versa. Sample splitting avoids the Donsker conditions that were prevalent in the classic semi-parametric inference literature and only requires a root-mean-squared-error (RMSE) guarantee of the estimate $\hat{g}$ of $o_p(n^{-1/4})$.

However, sample splitting or cross-fitting still leads to poorer sample usage, as we can lose half of our data when training complex non-parametric or machine learning models, which can be problematic in small and moderate sample regimes. Our main result is to show that root-$n$ consistency and asymptotic normality of the standard algorithm, without sample splitting, can be achieved without the Donsker property, but solely if one assumes that the first stage estimation algorithm is $o(n^{-1/2})$ leave-one-out stable, a relatively widely studied property in the statistical machine learning and generalization theory literature [15, 34, 23, 25]. As a leading example we show that our stability conditions are satisfied by bagging estimators formed with sub-sampling without replacement.

Recent prior work of [21] also considered asymptotic normality based on stability conditions, but as we expand in the main text, their requirement on the stability property is much harsher than the one we derive here. Moreover, [21] analyzes only a special case of the class of moment problems that we consider here. For instance, in our leading example of bagging estimators, the prior result of [21] would require that the bias of the base estimator decays faster than $1/n$, where $n$ is the sample size, which is typically not the case. In contrast, our stability condition does not impose any explicit assumption on the bias and solely requires that the sub-sample size $m$ is $o(\sqrt{n})$.

To simplify the regularity assumptions required for asymptotic normality, we focus on the case where $m(Z; \theta, g)$ is linear in $\theta$, i.e.

$$m(Z; \theta, g) = a(Z; g)\,\theta + \nu(Z; g)$$

where $a(Z; g) \in \mathbb{R}^{p \times p}$ is a $p \times p$ matrix and $\nu(Z; g) \in \mathbb{R}^p$ is a $p$-vector, and we denote with:

$$A(g) := \mathbb{E}_Z[a(Z; g)] \qquad\qquad A_n(g) := \mathbb{E}_n[a(Z; g)]$$
$$V(g) := \mathbb{E}_Z[\nu(Z; g)] \qquad\qquad V_n(g) := \mathbb{E}_n[\nu(Z; g)].$$

Many of the leading examples in semi-parametric problems that arise in causal inference correspond to linear moment problems. We present below a representative set of problems that are widely used in the practice of causal inference.

**Example 1** (Partially Linear Treatment Effect [51]). *If one assumes that the outcome of interest $Y$ is linear in the treatment, i.e. $Y = \theta_0' T + f_0(X) + \epsilon$, with $\mathbb{E}[\epsilon \mid T, X] = 0$, then estimating the treatment effect $\theta_0$ boils down to solving the following linear moment:*

$$m(Z; \theta, g) = (Y - q(X) - \theta'(T - p(X)))\,(T - p(X))$$

*where the corresponding true values of $(q, p)$ are $q_0(X) = \mathbb{E}[Y|X]$, $p_0(X) = \mathbb{E}[T|X]$.*

**Example 2** (Partially Linear IV [17]). *If one assumes that the outcome of interest $Y$ is linear in the treatment, i.e. $Y = \theta_0' T + f_0(X) + \epsilon$, but the treatment is endogenous (i.e. there are unobserved confounders) and one has access to a random variable $Z$, that is referred to as an instrument, which correlates with the treatment but is un-correlated with the residual in the outcome equation, i.e. satisfies that $\mathbb{E}[\epsilon \mid Z, X] = 0$, then estimating $\theta_0$ boils down to solving the following linear moment:*

$$m(Z; \theta, g) = (Y - q(X) - \theta'(T - p(X)))\,(Z - r(X))$$

*where the true values of $(q, p, r)$ are $q_0(X) = \mathbb{E}[Y|X]$, $p_0(X) = \mathbb{E}[T|X]$, $r_0(X) = \mathbb{E}[Z|X]$.*

**Example 3** (Average linear functionals of regression functions). *Consider a class of moment functions of the form:*

$$m(Z; \theta, g) = \theta - m_b(Z; q) - \mu(T, X)(Y - q(T, X))$$

*where $g = (q, \mu)$, $m_b$ is a linear functional of $q$ and the corresponding true value $q$ is a regression function $q_0(T, X) = \mathbb{E}[Y \mid T, X]$ and $\mu_0$ is Riesz representer of the functional $\mathbb{E}[m_b(Z; q)]$ (see [20] for examples). For instance, in the case of a binary treatment $T$, where we have that $Y = q(T, X) + \epsilon$ and $\mathbb{E}[\epsilon \mid T, X] = 0$, then the average treatment effect $\theta_0 = \mathbb{E}[q(1, X) - q(0, X)]$ is identified by a moment of the latter type, with:*

$$m_b(Z; q) = q(1, X) - q(0, X) \qquad \text{(Average Treatment Effect (ATE))}$$

*While if we have a target treatment policy $\pi : \mathcal{X} \to \{0, 1\}$, and we want to estimate its average value $\theta_0$, we can identify do so with a moment of the aforementioned type, with:*

$$m_b(Z; q) = \pi(X)(q(1, X) - q(0, X)) \qquad \text{(Average Policy Effect)}$$

For completeness, we also include in Appendix H an extension of our results to nonlinear moment problems.

## 2   Asymptotic Normality without Sample Splitting

We start by providing an asymptotic normality theorem for semi-parametric moment estimators without sample splitting and where the moment satisfies the well-studied property of Neyman orthogonality. Our theorem requires four main conditions: i) root-mean-squared-error (RMSE) rates for the nuisance function estimates of $o_p(n^{-1/4})$, ii) Neyman orthogonality of the moment with respect to the nuisances, iii) second-order smoothness of the moment with respect to the nuisance functions and, iv) stochastic equicontinuity of the Jacobian and the offset part of the linear moment function as the nuisance estimate $\hat{g}$ converges to $g_0$. For a vector $x \in \mathbb{R}^p$ we denote with $\|x\|_2$ the $\ell_2$ norm and for a matrix $X \in \mathbb{R}^{p \times p}$ we denote with $\|X\|_{op}$ the operator norm with respect to the $\ell_2$ norm.

**Theorem 1.** *Suppose that the nuisance estimate $\hat{g} \in \mathcal{G}$ satisfies:*

$$\|\hat{g} - g_0\|_2^2 \triangleq \mathbb{E}_X\left[\|\hat{g}(X) - g_0(X)\|_2^2\right] = o_p\left(n^{-1/2}\right). \qquad \text{(Consistency Rate)}$$

*Suppose that the moment satisfies the Neyman orthogonality condition: for all $g \in \mathcal{G}$*

$$D_g M(\theta_0, g_0)[g - g_0] \triangleq \frac{\partial}{\partial t} M(\theta_0, g_0 + t(g - g_0))\big|_{t=0} = 0 \qquad \text{(Neyman Orthogonality)}$$

*and a second-order smoothness condition: for all $g \in \mathcal{G}$*

$$D_{gg} M(\theta_0, g_0)[g - g_0] \triangleq \frac{\partial^2}{\partial t^2} M(\theta_0, g_0 + t(g - g_0))\big|_{t=0} = O\left(\|g - g_0\|_2^2\right) \qquad \text{(Smoothness)}$$

*and that the moment $m$ satisfy the stochastic equicontinuity conditions:*

$$\sqrt{n}\,\|A(\hat{g}) - A(g_0) - (A_n(\hat{g}) - A_n(g_0))\|_{op} = o_p(1)$$
$$\sqrt{n}\,\|V(\hat{g}) - V(g_0) - (V_n(\hat{g}) - V_n(g_0))\|_2 = o_p(1). \qquad \text{(Stochastic Equicontinuity)}$$

*Assume that $A(g_0)^{-1}$ exists and that for any $g, g' \in \mathcal{G}$:*

$$\|A(g) - A(g')\|_{op} = O\left(\|g - g'\|_2\right).$$

*Moreover, assume that for any $i, j \in [p] \times [p]$, the random variable $a_{i,j}(Z; g_0)$ has bounded variance and that $\|\theta_0\|_2 = O(1)$. Then $\hat{\theta}$ is asymptotically normal:*

$$\sqrt{n}\left(\hat{\theta} - \theta_0\right) \xrightarrow{n \to \infty, d} N\left(0, A(g_0)^{-1}\mathbb{E}\left[m(Z; \theta_0, g_0)\,m(Z; \theta_0, g_0)^\top\right]A(g_0)^{-1}\right).$$

The first three conditions are standard assumptions in the literature on debiased machine learning. The final condition (stochastic equicontinuity) is exactly where sample splitting comes very handy in the literature. To illustrate the reason why sample splitting helps with the stochastic equicontinuity condition, let us consider the first part of the condition (the reasoning is analogous for the second part). It asks that the difference of two centered empirical processes, namely $A_n(\hat{g}) - A(\hat{g})$ and $A_n(g_0) - A(g_0)$, goes to zero faster than $n^{-1/2}$. We expect each empirical process to go down to zero at exactly $n^{-1/2}$ and so this condition asks, since $\hat{g}$ converges to $g_0$ is the empirical process continuous in its argument and for that reason does the difference converge to zero faster than each individual component. If the estimate $\hat{g}$ was fitted on a separate sample, then conditional on $\hat{g}$, we have that each element $t = (i,j) \in [p] \times [p]$ of $A_n(\hat{g}) - A_n(g_0)$ is an empirical average of iid random variables with mean $A(\hat{g}) - A(g_0)$. Thus a simple Bernstein inequality would show that the difference of the two empirical processes would converge to zero at the order of:

$$O_p\left(\sqrt{\frac{\mathbb{E}[(a_t(Z;\hat{g}) - a_t(Z;g_0))^2]}{n}} + \frac{1}{n}\right) = O_p\left(\sqrt{\frac{\|\hat{g} - g_0\|_2^2}{n}} + \frac{1}{n}\right) = o_p(n^{-1/2})$$

where we also invoked a mean-squared-continuity property of $a_t(Z;g)$ and the fact that $\|\hat{g} - g_0\|_2 = o_p(1)$. Thus, with sample splitting, no further constraint is required from $\hat{g}$, other than a convergence rate on $\|\hat{g} - g_0\|_2$. In fact, as was noted in recent work of [22], in the above step it suffices to assume that $\mathbb{E}[(a_t(Z;\hat{g}) - a_t(Z;g_0))^2] = O\left(\|\hat{g} - g_0\|_2^q\right)$ for any $q < \infty$, which is a much weaker mean-squared-continuity assumption, and the property would still hold, since $\|\hat{g} - g_0\|_2^{q/2} n^{-1/2} = o(n^{-1/2})$, whenever $\|\hat{g} - g_0\|_2 = o_p(1)$.

Without sample splitting, note that $\hat{g}$ is now correlated with the samples in the empirical averages and hence $A_n(\hat{g}) - A_n(g_0)$ is no longer an average of i.i.d. random variables. Typical approaches would try to prove a uniform stochastic equicontinuity property over the function space $\mathcal{G}$, typically referred to as a Donsker property of the function space $\mathcal{G}$. In particular, if we could show that w.h.p.:

$$\forall g \in \mathcal{G} : O\left(\sqrt{\frac{\mathbb{E}[(a_t(Z;g) - a_t(Z;g_0))^2]}{n}} + \frac{1}{n}\right) = O\left(\delta_n\|g - g_0\|_2 + \delta_n^2\right) = O\left(\delta_n^2 + \|g - g_0\|_2^2\right)$$

then the above property would also hold for $\hat{g}$. Subsequently, since we know that $\|\hat{g} - g_0\|_2^2 = o_p(n^{-1/2})$, by our convergence rate assumptions on $\hat{g}$, then it would suffice that $\delta_n^2 = o(n^{-1/2})$. Such localized concentration inequalities have been known to hold for Donsker classes, which are typically defined via entropy integrals, and more recently it was also noted that such inequalities are satisfied with $\delta_n$ being the critical radius of the space $\mathcal{G}$, defined via localized Rademacher complexities.

However, the latter approach is conservative as it requires a uniform control over the function space $\mathcal{G}$ and does not utilize at all the properties of the estimation algorithm itself. In particular, as we will show in the next section, the main result of our work is that this stochastic equicontinuity condition follows from $o(n^{-1/2})$ leave-one-out stability conditions on our estimation algorithm, which are typical in the machine learning literature and in the excess risk and generalization bounds literature.

## 3 Stochastic Equicontinuity via Stability

We will show that the Condition (Stochastic Equicontinuity) is satisfied, whenever the estimate $\hat{g}$ satisfies leave-one-out stability properties and the moment satisfies the weak mean-squared-continuity property of [22]. We start by some preliminary definitions required to state our stability conditions. Define $Z^{(-l)}$ as the data $Z_1, \ldots, Z_n$ with the $l$-th data point $Z_l$ replaced with an independent copy $\tilde{Z}_l$. Define $Z^{(-l_1,-l_2)}$ as the data $Z_1, \ldots, Z_n$ with both the $l_1$-th and the $l_2$-th data points $Z_{l_1}, Z_{l_2}$ replaced with independent copies $\tilde{Z}_{l_1}, \tilde{Z}_{l_2}$. Define similarly for $Z^{(-l_1,-l_2,-l_3)}$ and so on. Let $\hat{g}^{(-l)}$ be the estimator trained on $Z^{(-l)}$ instead of $Z_1, \ldots, Z_n$. Similarly, let $\hat{g}^{(-l_1,-l_2)}$ be trained on $Z^{(-l_1,-l_2)}$ and so on. Moreover, we will always denote with $Z$ a fresh random variable drawn from the distribution of the samples, but which is not part of any training sample. For any random variable $X$, we denote with $\|X\|_1 := \mathbb{E}[|X|]$, with $\|X\|_2 := \sqrt{\mathbb{E}[X^2]}$, and with $\|X\|_p := (\mathbb{E}[|X|^p])^{1/p}$ for any $p \geq 1$ in general.

**Lemma 2** (Main Lemma). *If the estimation algorithm satisfies the stability conditions: for all* $i, j \in [p]$

$$\max_{l \in [n]} \left\| a_{i,j}(Z_l, \hat{g}) - a_{i,j}(Z_l, \hat{g}^{(-l)}) \right\|_1 = o(n^{-1/2}) \quad \max_{l \in [n]} \left\| a_{i,j}(Z, \hat{g}) - a_{i,j}(Z, \hat{g}^{(-l)}) \right\|_2 = o(n^{-1/2})$$

$$\max_{l \in [n]} \left\| \nu_i(Z_l, \hat{g}) - \nu_i(Z_l, \hat{g}^{(-l)}) \right\|_1 = o(n^{-1/2}) \quad \max_{l \in [n]} \left\| \nu_i(Z, \hat{g}) - \nu_i(Z, \hat{g}^{(-l)}) \right\|_2 = o(n^{-1/2})$$

*and the moment satisfies the mean-squared-continuity condition:*

$$\forall g, g' : \mathbb{E}[(a_{i,j}(Z; g) - a_{i,j}(Z; g'))^2] \leq L\|g - g'\|_2^q \quad \mathbb{E}[(\nu_i(Z; g) - \nu_i(Z; g'))^2] \leq L\|g - g'\|_2^q$$

*for some* $0 < q < \infty$ *and some* $L > 0$*, then the Condition (*Stochastic Equicontinuity*) is satisfied.*

**Remark 1.** *We show in section 3.1 that the stability conditions are tight. We present a counter example for which* $\left\| \nu_i(Z, \hat{g}) - \nu_i(Z, \hat{g}^{(-l)}) \right\|_2$ *is exactly of order* $n^{-1/2}$ *and for which the Stochastic Equicontinuity condition is not satisfied.*

**Remark 2.** *We note that prior work of [21] that established asymptotic normality without sample splitting via stability, required significantly stronger conditions than what we invoke here. In particular, if we let* $\beta_n$ *be the stability of the estimator* $\hat{g}$ *as measured by the quantities in Lemma 2, then the prior work of [21], would require that* $n\beta_n\|\hat{g} - g_0\|_2 \to 0$. *If we only know that* $\|\hat{g} - g_0\|_2 = o(n^{-1/4})$, *then the above would require* $\beta_n = o(n^{-3/4})$, *which is much slower than* $o(n^{-1/2})$. *Moreover, for bagged kernel estimators that we analyze in section 4, the prior work would require that if we use bags of size* $m$, *then the bias of the base estimator with* $m$ *samples, denoted as* bias$(m)$ *satisfies that* $m$bias$(m) \to 0$. *This would rarely be satisfied and in prior work, the only concrete case that was given was forest estimators with binary variables under strong sparsity conditions, in which case the bias decays exponentially with the sample size. For more general estimators, we expect* bias$(m) = 1/m^\alpha$, *for some* $\alpha$. *For such settings, our work still applies and, as we show in section 4, gives results for bagged* 1*-nearest neighbor estimation algorithms, which do not satisfy any entropy or critical radius bound, but are stable. The key innovation that enables our improved results is a "double centering" approach that derives intuition from techniques invoked in the analysis of cross-validation via stability and the proof of the Efron-Stein inequality[14]. This idea has already been used in the study of the cross validated risk [4, 5].*

*Proof of Main Lemma.* We will show the first part of the lemma, i.e. that if for all $i, j \in [p]$

$$\max_{l \in [n]} \left\| a_{i,j}(Z_l, \hat{g}) - a_{i,j}(Z_l, \hat{g}^{(-l)}) \right\|_1 = o(n^{-1/2}) \quad \max_{l \in [n]} \left\| a_{i,j}(Z, \hat{g}) - a_{i,j}(Z, \hat{g}^{(-l)}) \right\|_2 = o(n^{-1/2})$$

then

$$\sqrt{n} \left\| A(\hat{g}) - A(g_0) - (A_n(\hat{g}) - A_n(g_0)) \right\|_{op} = o_p(1)$$

The analogous statement for $\nu$ and $V$, follows in an identical manner.

Since $A(g)$ and $A_n(g)$ are $p \times p$ matrices and $p = O(1)$, it suffices to show the above property for every element $(i, j) \in [p] \times [p]$, i.e. that

$$\sqrt{n} \left| A_{i,j}(\hat{g}) - A_{i,j}(g_0) - (A_{n,i,j}(\hat{g}) - A_{n,i,j}(g_0)) \right| = o_p(1).$$

For this it suffices to show that:

$$J_n := \sqrt{n} \left\| A_{i,j}(\hat{g}) - A_{i,j}(g_0) - (A_{n,i,j}(\hat{g}) - A_{n,i,j}(g_0)) \right\|_1 = o(1).$$

**In the remainder of the proof we look at a particular** $(i, j)$ **and hence for simplicity we overload notation and we let** $a := a_{i,j}$ **and** $A := A_{i,j}$**.**

By triangle inequality and domination of $L^p$ norms we have

$$J_n = \left\| \frac{1}{\sqrt{n}} \sum_{l=1}^{n} \left\{ a(Z_l, \hat{g}) - A(\hat{g}) - \left[ a(Z_l, g_0) - A(g_0) \right] \right\} \right\|_1$$

$$\leq \left\| \frac{1}{\sqrt{n}} \sum_{l=1}^{n} \left\{ a(Z_l, \hat{g}) - a(Z_l, \hat{g}^{(-l)}) \right\} \right\|_1 + \sqrt{n} \max_{l \in [n]} \left\| A(\hat{g}) - A(\hat{g}^{(-l)}) \right\|_1$$

$$+ \left\| \frac{1}{\sqrt{n}} \sum_{l=1}^{n} \left\{ a(Z_l, \hat{g}^{(-l)}) - a(Z_l, g_0) - \left( A(\hat{g}^{(-l)}) - A(g_0) \right) \right\} \right\|_2.$$

To ease notations, we denote

$$J_{1,n} := \left\| \frac{1}{\sqrt{n}} \sum_{l=1}^{n} \left\{ a(Z_l, \hat{g}) - a(Z_l, \hat{g}^{(-l)}) \right\} \right\|_1,$$

$$J_{2,n} := \sqrt{n} \max_{l \in [n]} \left\| A(\hat{g}) - A(\hat{g}^{(-l)}) \right\|_1,$$

$$J_{3,n} := \left\| \frac{1}{\sqrt{n}} \sum_{l} \left\{ a(Z_l, \hat{g}^{(-l)}) - a(Z_l, g_0) - \left( A(\hat{g}^{(-l)}) - A(g_0) \right) \right\} \right\|_2.$$

Now we have that by triangle inequality

$$J_{1,n} \leq \frac{1}{\sqrt{n}} \sum_{l} \left\| a(Z_l, \hat{g}) - a(Z_l, \hat{g}^{(-l)}) \right\|_1 \leq \sqrt{n} \max_{\ell \in [n]} \left\| a(Z_l, \hat{g}) - a(Z_l, \hat{g}^{(-l)}) \right\|_1 = o(1).$$

Similarly we can handle $J_{2,n}$:

$$J_{2,n} \leq \sqrt{n} \max_{l \in [n]} \left\| a(Z, \hat{g}) - a(Z, \hat{g}^{(-l)}) \right\|_1$$

$$\leq \sqrt{n} \max_{l \in [n]} \left\| a(Z, \hat{g}) - a(Z, \hat{g}^{(-l)}) \right\|_2 = o(1).$$

We now aim to show that $J_{3,n} = o(1)$. Write for simplicity

$$K_l := a(Z_l, \hat{g}^{(-l)}) - a(Z_l, g_0) - \left( A(\hat{g}^{(-l)}) - A(g_0) \right)$$

Now by expanding the square we obtain

$$J_{3,n}^2 = \mathbb{E}\left[ \left( \frac{1}{\sqrt{n}} \sum_{l=1}^{n} K_l \right)^2 \right] \leq \max_{l \in [n]} \mathbb{E}\left[ K_l^2 \right] + (n-1) \max_{l_1 \neq l_2 \in [n]} \mathbb{E}[K_{l_1} K_{l_2}]$$

To bound the first term, we have by mean-squared continuity:

$$\mathbb{E}[K_l^2] = \mathbb{E}\left[ \left( a(Z_l, \hat{g}^{(-l)}) - a(Z_l, g_0) - \left( A(\hat{g}^{(-l)}) - A(g_0) \right) \right)^2 \right]$$

$$\leq 2\mathbb{E}\left[ \left( a(Z_l, \hat{g}^{(-l)}) - a(Z_l, g_0) \right)^2 \right] + 2\mathbb{E}\left[ \left( A(\hat{g}^{(-l)}) - A(g_0) \right)^2 \right]$$

$$= 2\mathbb{E}\left[ \mathbb{E}\left[ \left( a(Z_l, \hat{g}^{(-l)}) - a(Z_l, g_0) \right)^2 \Big| Z^{(-l)} \right] \right] + 2\mathbb{E}\left[ \mathbb{E}\left[ \left( A(\hat{g}^{(-l)}) - A(g_0) \right)^2 \Big| Z^{(-l)} \right] \right]$$

$$\leq 2L \cdot \mathbb{E}\left[ \left\| \hat{g}^{(-l)} - g_0 \right\|_2^q \right] + 2L \cdot \mathbb{E}\left[ \left\| \hat{g}^{(-l)} - g_0 \right\|_2^q \right] = 4L \cdot \mathbb{E}\left[ \|\hat{g} - g_0\|_2^q \right] = o(1).$$

where we invoked the property that $\|\hat{g} - g_0\|_2 = o_p(1)$, and the second to last equality exploited the tower law. Thus $\max_{l \in [n]} \mathbb{E}\left[ K_l^2 \right] = o(1)$.

**Double centering.** We now bound the term, $(n-1) \max_{l_1 \neq l_2 \in [n]} \mathbb{E}[K_{l_1} K_{l_2}]$. Define, for simplicity, for $l_1 \neq l_2$

$$K_{l_1}^{(l_2)} := a(Z_{l_1}, \hat{g}^{(-l_1, -l_2)}) - a(Z_{l_1}, g_0) - \left( A(\hat{g}^{(-l_1, -l_2)}) - A(g_0) \right).$$

Note that $K_{l_1}^{(l_2)}$ does not depend on the $l_2$-th data point and is only a function of $Z^{(-l_2)}, \tilde{Z}_{l_1}$. Moreover, by the definition of $A$, noting that $Z_l$ is independent of $Z^{(-l)}$ and $Z_{l_1}$ is independent of $Z^{(-l_1, -l_2)}$:

$$\mathbb{E}\left[ K_l \Big| Z^{(-l)} \right] = \mathbb{E}\left[ a(Z_l, \hat{g}^{(-l)}) - A(\hat{g}^{(-l)}) \Big| Z^{(-l)} \right] - \mathbb{E}\left[ a(Z_l, g_0) - A(g_0) \Big| Z^{(-l)} \right]$$

$$= \mathbb{E}\left[ a(Z_l, \hat{g}^{(-l)}) - A(\hat{g}^{(-l)}) \Big| Z^{(-l)} \right] - \mathbb{E}\left[ a(Z_l, g_0) - A(g_0) \right] = 0$$

and

$$\mathbb{E}\left[K_{l_1}^{(l_2)}\Big|Z^{(-l_1,-l_2)}\right]$$

$$= \mathbb{E}\left[a(Z_{l_1},\hat{g}^{(-l_1,-l_2)}) - A(\hat{g}^{(-l_1,-l_2)})\Big|Z^{(-l_1,-l_2)}\right] + \mathbb{E}\left[a(Z_{l_1},g_0) - A(g_0)\Big|Z^{(-l_1,-l_2)}\right]$$

$$= \mathbb{E}\left[a(Z_{l_1},\hat{g}^{(-l_1,-l_2)}) - A(\hat{g}^{(-l_1,-l_2)})\Big|Z^{(-l_1,-l_2)}\right] + \mathbb{E}\left[a(Z_{l_1},g_0) - A(g_0)\right] = 0$$

For simplicity, we show that $(n-1)\mathbb{E}[K_1 K_2] = o(1)$, i.e. we show that the second term vanishes for $l_1 = 1$ and $l_2 = 2$. The same exact arguments generalize to arbitrary $l_1, l_2$. We begin by writing:

$$(n-1)\mathbb{E}[K_1 K_2] = (n-1)\mathbb{E}\left[\left(K_1 - K_1^{(2)}\right)K_2\right],$$

since by tower law:

$$\mathbb{E}\left[K_1^{(2)}K_2\right] = \mathbb{E}\left[\mathbb{E}\left[K_2\Big|Z^{(-2)},\tilde{Z}_1\right]K_1^{(2)}\right] = \mathbb{E}\left[\mathbb{E}\left[K_2\Big|Z^{(-2)}\right]K_1^{(2)}\right] = 0$$

Similarly by conditioning on $Z^{(-1)}, \tilde{Z}_2$ and using tower law, we can show that

$$\mathbb{E}\left[\left(K_1 - K_1^{(2)}\right)K_2^{(1)}\right] = \mathbb{E}\left[\mathbb{E}\left[K_1 - K_1^{(2)}\Big|Z^{(-1)},\tilde{Z}_2\right]K_2^{(1)}\right]$$

$$= \mathbb{E}\left[\left\{\mathbb{E}\left[K_1\Big|Z^{(-1)},\tilde{Z}_2\right] - \mathbb{E}\left[K_1^{(2)}\Big|Z^{(-1)},\tilde{Z}_2\right]\right\}K_2^{(1)}\right]$$

$$= \mathbb{E}\left[\left\{\mathbb{E}\left[K_1\Big|Z^{(-1)}\right] - \mathbb{E}\left[K_1^{(2)}\Big|Z^{(-1,-2)}\right]\right\}K_2^{(1)}\right] = 0.$$

Hence, we have

$$(n-1)\mathbb{E}[K_1 K_2] = (n-1)\mathbb{E}\left[\left(K_1 - K_1^{(2)}\right)K_2\right] = (n-1)\mathbb{E}\left[\left(K_1 - K_1^{(2)}\right)\left(K_2 - K_2^{(1)}\right)\right]$$

With identical arguments, the same equality holds for any indices $l_1, l_2$. By Cauchy-Schwarz:

$$\max_{l_1 \neq l_2}(n-1)\mathbb{E}[K_{l_1}K_{l_2}] = \max_{l_1 \neq l_2}(n-1)\mathbb{E}\left[\left(K_{l_1} - K_{l_1}^{(l_2)}\right)\left(K_{l_2} - K_{l_2}^{(l_1)}\right)\right]$$

$$\leq \max_{l_1 \neq l_2}(n-1)\left\|K_{l_1} - K_{l_1}^{(l_2)}\right\|_2\left\|K_{l_2} - K_{l_2}^{(l_1)}\right\|_2$$

Thus for $\max_{l_1 \neq l_2}(n-1)\mathbb{E}[K_{l_1}K_{l_2}] = o(1)$ it suffices that: $\max_{l_1 \neq l_2}\left\|K_{l_1} - K_{l_1}^{(l_2)}\right\|_2 = o(n^{-1/2})$. Expanding the definitions $K_{l_1}$ and $K_{l_1}^{(l_2)}$, the above simplifies to:

$$\left\|K_{l_1} - K_{l_1}^{(l_2)}\right\|_2 = \left\|a(Z_{l_1},\hat{g}^{(-l_1)}) - A(\hat{g}^{(-l_1)}) - \left(a(Z_{l_1},\hat{g}^{(-l_1,-l_2)}) - A(\hat{g}^{(-l_1,-l_2)})\right)\right\|_2.$$

The latter is upper bounded by a triangle inequality and a Jensen's inequality by:

$$\left\|K_{l_1} - K_{l_1}^{(l_2)}\right\|_2 \leq 2\left\|a(Z_{l_1},\hat{g}^{(-l_1)}) - a(Z_{l_1},\hat{g}^{(-l_1,-l_2)})\right\|_2.$$

If we denote with $Z$ a fresh random sample not part of the training sets, then since $Z_{l_1}$ is not part of $Z^{(-l_1)}$, we have:

$$\left\|a(Z_{l_1},\hat{g}^{(-l_1)}) - a(Z_{l_1},\hat{g}^{(-l_1,-l_2)})\right\|_2 = \left\|a(Z,\hat{g}^{(-l_1)}) - a(Z,\hat{g}^{(-l_1,-l_2)})\right\|_2$$

$$= \left\|a(Z,\hat{g}) - a(Z,\hat{g}^{(-l_2)})\right\|_2$$

where we also used the fact that $\tilde{Z}_{l_1}$ only appears in the training sets of $\hat{g}^{(-l_1)}$ and $\hat{g}^{(-l_1,-l_2)}$ and we can simply rename it to $Z_{l_1}$, as they are identically distributed and both independent from all the other data points. Invoking the second of our stability conditions for $a$, we have that: $\max_{l_1 \neq l_2}\left\|K_{l_1} - K_{l_1}^{(l_2)}\right\|_2 = o(n^{-1/2})$. Hence, $J_{3,n} = o(1)$, which completes the proof. $\qquad\square$

## 3.1 Tightness of Stability Condition

We present here an example that shows that, without further structural constraints (on the moment or the function space $\mathcal{G}$), the stability condition we impose is required for stochastic equicontinuity condition to hold. Let $(X_i) \overset{i.i.d}{\sim} \text{unif}[0,1]$ be i.i.d uniform random variables. We set $Y_i := \mathbb{I}(X_i \le 0.5)$ and set $Z_i := (X_i, Y_i)$. For any $x \in [0,1]$, we define $c(Z_{1:n}, x) := \arg\min_{i \le n} |X_i - x|$ the function that returns the index of the nearest example to $x$ in $\{X_1, \ldots, X_n\}$ and note that the quantity $Y_{c(Z_{1:n},x)}$ is its nearest neighbour estimator. Let

$$\hat{g}(Z_{1:n})(x,y) := n^{1/6}\mathbb{I}(y \ne Y_{c(Z_{1:n},x)}), \qquad \nu(Z,g) := g(Z)^3.$$

We remark that $\nu(\cdot, \hat{g})$ does not satisfy our stability conditions as $\|\nu(Z,\hat{g}) - \nu(Z,\hat{g}^{(-1)})\|_2$ is exactly of order $1/\sqrt{n}$, neither does it respect the stochastic equicontinuity property.

**Lemma 3.** *Let $(X_i) \overset{i.i.d}{\sim} \text{unif}[0,1]$ be i.i.d uniform random variables. We set $Y_i := \mathbb{I}(X_i \le 0.5)$ and set $Z_i := (X_i, Y_i)$. There are constants $C, c > 0$ such that*

- *Set $g_0(Z) := 0$ then we have $\|\hat{g}(Z) - g_0(Z)\|_2^2 \to 0$*

- $\frac{C}{\sqrt{n}} \ge \|\nu(Z,\hat{g}) - \nu(Z,\hat{g}^{(-1)})\|_2 \ge \frac{c}{\sqrt{n}}$

- $\sqrt{n}|V(\hat{g}) - V(g_0) - (V_n(\hat{g}) - V_n(g_0))| \not\overset{P}{\to} 0.$

# 4 Application: Bagging Estimators

We remark that if the functions $a$ and $\nu$ satisfy certain $L^p$-Lipschitz conditions then the stability conditions of lemma 2 are implied by the algorithmic stability of the estimator $\hat{g}$. Those conditions are only marginally stronger than the condition of mean-squared-continuity found in lemma 2.

**Corollary 4.** *Fix any constant $r > 1$. Suppose that there is $L < \infty$ such that the estimation algorithm satisfies the following uniform $L^{2r}$-continuity condition: for all $i,j \in [p]$ and $l \in [n]$*

$$\mathbb{E}[(a_{i,j}(Z_l; \hat{g}) - a_{i,j}(Z_l; \hat{g}^{(-l)}))^2] \le L \cdot \mathbb{E}\Big[\sup_x \|\hat{g}(x) - \hat{g}^{(-l)}(x)\|_2^{2r}\Big]^{1/r} \tag{1}$$

$$\mathbb{E}[(\nu_i(Z_l; \hat{g}) - \nu_i(Z_l; \hat{g}^{(-l)}))^2] \le L \cdot \mathbb{E}\Big[\sup_x \|\hat{g}(x) - \hat{g}^{(-l)}(x)\|_2^{2r}\Big]^{1/r} \qquad (L^{2r}\text{-Continuity})$$

$$\mathbb{E}[(a_{i,j}(Z; \hat{g}) - a_{i,j}(Z; \hat{g}^{(-l)}))^2] \le L \cdot \mathbb{E}\Big[\sup_x \|\hat{g}(x) - \hat{g}^{(-l)}(x)\|_2^{2r}\Big]^{1/r}$$

$$\mathbb{E}[(\nu_i(Z; \hat{g}) - \nu_i(Z; \hat{g}^{(-l)}))^2] \le L \cdot \mathbb{E}\Big[\sup_x \|\hat{g}(x) - \hat{g}^{(-l)}(x)\|_2^{2r}\Big]^{1/r}.$$

*Suppose in addition that*

$$\max_{l \le n} \mathbb{E}_{Z_{1:n}}\Big[\sup_x \|\hat{g}(x) - \hat{g}^{(-l)}(x)\|_2^{2r}\Big]^{1/2r} = o\Big(n^{-1/2}\Big) \qquad (\text{Algorithmic Stability})$$

*and if in addition mean-squared-continuity condition is satisfied, then the conditions of lemma 2 are satisfied.*

The uniform $L^{2r}$-continuity conditions are going to be satisfied by most moment functions $m(\cdot; \cdot, \cdot)$. We notably show in the appendix that example 1, example 2 and example 3 with general moment conditions satisfy our conditions.

The condition of (Algorithmic Stability) is a commonly assumed condition in recent literature [4, 5] and is satisfied by various regularized empirical risk minimization estimators and stochastic gradient descent estimators [15, 24]. Notably, it is satisfied by bagged estimators. We show in theorem 5 that under very general conditions a bagged ensemble of any machine learning estimator is stable. Let $Z_1, \ldots, Z_n \in \mathcal{Z}$ be an independent and identically distributed (i.i.d.) sample of size $n$. We sample uniformly randomly without replacement from these observations repeatedly and independently, each time taking a sample of size $m$, for a number of $B$ times. We denote the resulting samples as

$$Z_{1:m}^b := \{Z_1^b, \ldots, Z_m^b\}, b \in [B].$$

Let $\hat{h}_m : \mathcal{Z}^m \to \mathbb{R}^P$ be a base machine learning estimator trained on $m$ observations. This base estimator can be tree, a CNN, a nearest-neighbour classifier, or any other type of machine learning estimator. The corresponding bagged estimator is

$$\hat{g}(\cdot) = \frac{1}{B} \sum_{b=1}^{B} \hat{h}_m(Z_{1:m}^b)(\cdot).$$

We will show that subject to $B$ and $m$ being sufficiently big and mild moment conditions on the base estimator $\hat{h}_m$, the bagged estimator satisfies condition (Algorithmic Stability).

**Theorem 5.** *Fix any constants $s, k \geq 2r$ such that $\frac{1}{s} + \frac{1}{k} = \frac{1}{2r}$. Assume $B, m$ are sufficiently large:*

$$m = o(\sqrt{n}) \qquad B >> m^{2/k} \cdot n^{1-\frac{2}{k}},$$

*and assume the base estimator $\hat{h}$ has bounded moments:*

$$\max_{l \leq n} \left\| \sup_x \left\| \hat{h}_m(Z_{1:m}^1)(x) \right\|_2 \right\|_s \leq C$$

*for some constant $C > 0$. Then (Algorithmic Stability) is achieved:*

$$\max_{l \leq n} \left\| \sup_x \| \hat{g}(x) - \hat{g}^{(-l)}(x) \|_2 \right\|_{2r} = o(n^{-1/2}).$$

*Therefore if $a$ and $\nu$ satisfy the condition (1) then the condition (2) is satisfied.*

In particular, we can take the base machine learning estimator to be the 1-nearest neighbor estimator. This specific choice leads to a bagged estimator that can be proved to satisfy both our stability conditions and our consistency rate condition in Theorem 1, in regression settings where covariates are of small intrinsic dimension.

Let $\{Z_i = (X_i, Y_i)\}_{i=1}^n$ be a sample of size $n$, drawn independently and identically distributed from $Z = (X, Y)$. Here $X \in \mathcal{X} \subset \mathbb{R}^D$ are known covariates and $Y \in \mathcal{Y} \subset \mathbb{R}^P$ is the response variable. As in our bagging setting, let $Z_{1:m}^1, \ldots, Z_{1:m}^B$ be B independent samples of size $m$ drawn without replacement from observations $\{Z_i\}_{i=1}^n$. A bagged 1-nearest neighbor (1-NN) estimator takes the following form:

$$\hat{g}(x) := \frac{1}{B} \sum_{b=1}^{B} \sum_{i=1}^{n} \mathbb{1}_{\{X_i = S_b(x)\}} Y_i,$$

where $S_b(x)$ is the 1-NN of $x$ in the set $Z_{1:m}^b$. The estimator is used to estimate the conditional expectation $g_0(x) := \mathbb{E}[Y \mid X = x]$.

**Lemma 6** (Special Case of Theorem 3 of [35]). *Assume that*

- *The marginal distribution $\mu$ of $X_1$ satisfies $(C, d)$-homogeneity on the ball $B(x, r)$:*

$$\mu(B(x, r)) \leq C\alpha^{-d} \mu(B(x, \alpha r)) \qquad \forall \alpha \in (0, 1)$$

  *for some $C, r > 0$. Here $d$ is referred to as the intrinsic dimension of the distribution.*

- *The conditional expectation $\mathbb{E}[Y \mid X = x]$ is a Lipschitz function in the coordinates $x$.*

- *The response variable $Y$ is bounded in $L^\infty$-norm: $\|Y\|_\infty < \infty$.*

*Then the bagged 1-NN estimator $\hat{g}$ with $B \geq \frac{n}{m}$ satisfies the following convergence condition:*

$$\sqrt{\mathbb{E}\left[ \|\hat{g}(X) - g_0(X)\|_2^2 \right]} \leq O\left(m^{-1/d}\right) + O\left(\sqrt{\frac{mP \log\log(Pn/m)}{n}}\right).$$

In particular, we note that when $0 < d < 2$ and $m = O(n^{\frac{1}{2}-\epsilon})$ for some $0 < \epsilon \leq \frac{1}{2} - \frac{1}{4}d$, we immediately have that

$$\sqrt{\mathbb{E}\left[ \|\hat{g}(X) - g_0(X)\|_2^2 \right]} = o(n^{-1/4}),$$

achieving the convergence rate assumed in Theorem 1. Moreover, provided that we choose $B$ to be sufficiently large such that

$$B >> n^{\epsilon+\frac{1}{2}} \qquad B \geq n^{1-\frac{2\epsilon+1}{k}},$$

the required stability conditions can also be satisfied.

## 5 Experimental Evaluation

We consider a synthetic experimental evaluation of our main theoretical findings. We focus on the partially linear model with a scalar outcome $Y \in \mathbb{R}$, a scalar continuous treatment $T \in \mathbb{R}$ and many controls $X \in \mathbb{R}^{n_x}$, where: $T = p_0(X) + \eta$, $\eta \sim N(0,1)$, $Y = \theta_0 T + f_0(X) + \epsilon$, $\epsilon \sim N(0,1)$. Our goal is the estimation of the treatment effect $\theta_0$, while estimating the nuisance functions $p_0(X)$ and $q_0(X) = \theta_0 p_0(X) + f_0(X)$ in a flexible manner. We will consider estimation based on the orthogonal moment presented in Example 1. We considered sub-sampled 1-nearest neighbor regression (NN) and sub-sampled fully grown (with only 1 sample minimum leaf size) random forest (RF) regression (see Appendix A) for the estimation of $p_0$ and $q_0$, regressing $T$ on $X$ and $Y$ on $X$ correspondingly. The true functions $p_0$ and $f_0$ are actually linear in our data generating process, with $p_0(X) = \beta_0' X$ and $f_0(X) = \gamma_0' X$ and where $\beta_0$ and $\gamma_0$ have only one non-zero coefficient (1-sparse), and which is the same coefficient for both $\beta_0$ and $\gamma_0$. In other words, only one of the $n_x$ potential confounding variables $X$ is actually a confounder.

We evaluate the performance of the estimate for $\theta_0$ for a range of values of the sample size $n$ and the dimension of the controls $n_x$ and with or without cross-fitting. For the cross-fitted estimates we used 2 splits. For each specification we draw 1000 experimental samples to evaluate the distributional properties of the estimate. We considered sub-sample sizes for the nuisance regressions based on our theoretical $n^{0.49}$ specification and for larger specifications too. We find that the estimate without cross-fitting typically has almost equal bias and smaller (and always comparable) variance (due most probably to the smaller mean squared error of the nuisances, since they are trained on larger sizes), especially in smaller sample sizes, and has better coverage properties, when the sub-sample size is $m = o(n^{1/2})$. Moreover, the estimate is approximately normally distributed, even without cross-fitting as is verified qualitatively via quantile-quantile (Q-Q) plots (c.f. Appendix A).

We also report the mean of the estimate of the standard error across the 1000 experiments, to evaluate the bias in the estimation of the standard error. We find that the estimate of the standard error is more accurate without cross-fitting, potentially because the estimate of the standard error does not incorporate the extra variance that stems from the sample-splitting process. This inaccuracy of the standard error estimate is most probably the reason for the worst coverage properties of the confidence intervals with cross-fitting.

In summary, we verify experimentally that for stable estimators, with the theoretically required level of stability, sample splitting or cross-fitting is not needed to maintain asymptotic normality, small bias and nominal coverage.

| | | bias | std | std_est | cov95 | | | bias | std | std_est | cov95 |
|---|---|---|---|---|---|---|---|---|---|---|---|
| $n{=}50, n_x{=}1$ | cv=1 | 0.021 | 0.151 | 0.140 | 0.917 | $n{=}50, n_x{=}1$ | cv=1 | 0.008 | 0.186 | 0.135 | 0.826 |
| | cv=2 | 0.022 | 0.173 | 0.139 | 0.866 | | cv=2 | 0.011 | 0.196 | 0.136 | 0.811 |
| $n{=}50, n_x{=}2$ | cv=1 | 0.042 | 0.149 | 0.143 | 0.920 | $n{=}50, n_x{=}2$ | cv=1 | 0.011 | 0.193 | 0.139 | 0.834 |
| | cv=2 | 0.046 | 0.170 | 0.142 | 0.863 | | cv=2 | 0.020 | 0.196 | 0.138 | 0.831 |
| $n{=}100, n_x{=}1$ | cv=1 | 0.012 | 0.104 | 0.100 | 0.931 | $n{=}100, n_x{=}1$ | cv=1 | 0.001 | 0.132 | 0.098 | 0.838 |
| | cv=2 | 0.014 | 0.119 | 0.100 | 0.884 | | cv=2 | 0.004 | 0.143 | 0.098 | 0.812 |
| $n{=}100, n_x{=}2$ | cv=1 | 0.030 | 0.110 | 0.101 | 0.911 | $n{=}100, n_x{=}2$ | cv=1 | 0.003 | 0.135 | 0.098 | 0.824 |
| | cv=2 | 0.032 | 0.121 | 0.101 | 0.878 | | cv=2 | 0.011 | 0.141 | 0.099 | 0.828 |
| $n{=}500, n_x{=}1$ | cv=1 | 0.007 | 0.046 | 0.045 | 0.943 | $n{=}500, n_x{=}1$ | cv=1 | 0.002 | 0.056 | 0.045 | 0.881 |
| | cv=2 | 0.007 | 0.049 | 0.045 | 0.920 | | cv=2 | 0.002 | 0.062 | 0.045 | 0.829 |
| $n{=}500, n_x{=}2$ | cv=1 | 0.015 | 0.045 | 0.045 | 0.927 | $n{=}500, n_x{=}2$ | cv=1 | 0.003 | 0.056 | 0.045 | 0.878 |
| | cv=2 | 0.015 | 0.048 | 0.045 | 0.917 | | cv=2 | 0.003 | 0.060 | 0.045 | 0.851 |
| $n{=}1000, n_x{=}1$ | cv=1 | 0.003 | 0.032 | 0.032 | 0.946 | $n{=}1000, n_x{=}1$ | cv=1 | 0.001 | 0.039 | 0.032 | 0.883 |
| | cv=2 | 0.002 | 0.034 | 0.032 | 0.922 | | cv=2 | 0.000 | 0.043 | 0.032 | 0.849 |
| $n{=}1000, n_x{=}2$ | cv=1 | 0.010 | 0.031 | 0.032 | 0.944 | $n{=}1000, n_x{=}2$ | cv=1 | 0.001 | 0.039 | 0.032 | 0.891 |
| | cv=2 | 0.010 | 0.033 | 0.032 | 0.930 | | cv=2 | 0.000 | 0.043 | 0.032 | 0.853 |

(a) Sub-sampled 1-NN with $m = n^{0.49}$  (b) Sub-sampled 1-NN with $m = n^{10/11}$

Figure 1: Comparison of bias, variance and coverage properties, with (cv=2) and without (cv=1) cross-fitting (sample splitting), for the estimation of the treatment effect in the partially linear model, when a sub-sampled 1-NN estimation is used for the nuisance function estimation. $n$ is the number of samples and $n_x$ the number of controls.

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
