# OpenReview forum: "Debiased Machine Learning without Sample-Splitting for Stable Estimators"
_NeurIPS.cc/2022/Conference — NeurIPS 2022 Accept_

### Official Review · Reviewer_ZG2K · 2022-07-10

**Rating:** 6
**Confidence:** 4
**Soundness:** 4 excellent
**Presentation:** 4 excellent
**Contribution:** 4 excellent

**Summary:**

New conditions on nuisance function based on leave-one-out stability that warrant root-n consistency and asymptotic normality of target parameters are proposed and theoretically investigated. It is also concretely shown that these conditions are satisfied by ensemble bagged estimators.

**Questions:**

(1)	I believe this paper’s contribution is substantial. That being said, I wonder to what extent $m(Z;\theta,g)$’s linearity in $\theta$ can be relaxed. It would be great if some discussion along this vein could be included in the paper. If $\theta$ and $g$ are not separable, I guess the stochastic equicontinuity in line 102 will need to hold uniformly in $\theta$ for $\theta$ in a neighborhood of $\theta_0$? Would stability still be sufficient for stochastic equicontinuity? If $m(Z;\theta,g)$ is nonlinear in $\theta$, would linearization trick like Taylor expansion to first order work under some additional conditions?

(2)	In Theorem 5, is the norm $\|Y\|_s = (E[Y^s])^{1/s}$? If so, this norm is not defined.


**Limitations:**

The moment condition has to be linear in target parameters.

**Strengths And Weaknesses:**

Strengths:
+ New result can be applied on causal inference with machine learning method, which is timely and impactful
+ Theoretical merits are clearly demonstrated by comparing with relevant existing works
+ New mathematical arguments that lead to refined theoretical results, which are shown to be unimprovable

---

> ### Author Response · Authors · 2022-08-02
> **Answers to Reviewer ZG2K**
>
> Thank you very much for your input and effort in reviewing the paper. We appreciate your suggestions and comments.
>
> 1. About the linear moment assumption:
>
> Thanks for pointing this out. Yes, we agree that this is a restriction, and to alleviate this we have included an extension of our results to the nonlinear moment case (please see the new Appendix Section H for the revised version). In our extension, the stochastic equicontinuity condition just needs to hold for $\theta_0$, and stability conditions (when evaluated at $\theta_0$) still imply this stochastic equicontinuity condition. However, we will need slightly stronger uniform in $\theta$ stability conditions to establish consistency of $\hat\theta.$
>
> We omitted working out such an extension from the original submission for several reasons: 1) it introduces more technical notation and more technical regularity conditions, 2) it requires slightly stronger uniform in $\theta$ stability conditions for consistency, 3) it requires a separate technical consistency lemma, as is typical for nonlinear moments, 4) the asymptotic normality (root-n rate) result, as is standard for nonlinear moments, only starts to kick in once the estimate $\hat\theta$ is sufficiently close to $\theta_0$, which always happens after some sufficiently large sample size, but that sample size could be large and not well controlled. This is a typical behavior of nonlinear moment proofs, which makes them more brittle in finite samples. Moreover, we would also like to point out that in practice, most causal inference problems use linear moments [1, 2, 3]; and that many recent papers, including [4] and [5], impose even stronger restrictions on the moments functions. Most widely used in practice instantiations of the debiased machine learning (DML) framework involve linear moments. For instance, both the dml package in R and the econml package in python (used in industry) only implement linear moment cases of the DML framework.
>
> 2. About the definition of the $s$-norm in Theorem 5
>
> The $s$-norm in Theorem 5 is indeed defined in the way you mentioned. We will make sure to include a clear definition in the camera-ready version. Thanks for pointing it out.
>
>
> References:
>
> [1] Victor Chernozhukov, Denis Chetverikov, Mert Demirer, Esther Duflo, Christian Hansen, Whitney Newey, and
> James Robins. Double/debiased machine learning for treatment and structural parameters, 2018.
>
> [2] Victor Chernozhukov, W Newey, James Robins, and Rahul Singh. Double/de-biased machine learning of global
> and local parameters using regularized riesz representers. 2019.
>
> [3] Peter M Robinson. Root-n-consistent semiparametric regression. 1988.
>
> [4] Victor Chernozhukov, Whitney K. Newey, and Rahul Singh. A simple and general debiased machine learning theorem with finite sample guarantees. 2021.
>
> [5] Victor Chernozhukov, Whitney K. Newey, and Rahul Singh. Automatic debiased machine learning of causal and structural effects. 2022.

---

### Official Review · Reviewer_v7GP · 2022-07-11

**Rating:** 7
**Confidence:** 4
**Soundness:** 4 excellent
**Presentation:** 3 good
**Contribution:** 4 excellent

**Summary:**

The paper presents a theoretical argument that debiased machine learning does not require sample splitting when the estimation algorithms are leave-one-out stable.

**Questions:**

1. Could you explain in greater detail the level of difficulty expected in verifying the conditions of Lemma 2? What about compared to the level of difficulty verifying the conditions of Theorem 1? What about compared to conditions previously imposed in the literature? If the algorithmic stability conditions cannot be verified, what course of action would you recommend? Also, what precisely is the relationship between the "Algorithmic Stability" condition of Corollary 4 and the "Consistency Rate" condition of Theorem 1?

2. Could you explain
- the second to last equality in the unnumbered display equation following Line 183,
- the second to last equality in the unnumbered display equation following LIne 186, and
- the last equality in the unnumbered display equation following Line 198?

3. Could you explain in greater detail the significance of Lemma 3? In particular, how does this example illustrate that "the stability condition [you] impose is required for stochastic equicontinuity condition to hold" when it is later stated that "$\nu(\cdot, \hat g)$ does not satisfy our stability conditions"?

4. Would it be possible to observe experimentally the effect of algorithmic stability on asymptotic normality? For example, the size of $m$ is related to the stability of the bagged estimator. Can we observe the relationship of $m$ to the convergence to normal distribution by looking at a grid of $m$'s (either $m = an$ or $m = n^a$ for different values of $a$)?

5. I feel like the paper could use more editorial judgment. This is probably just my impression, but the current version feels like it used to be a longer paper that was abruptly cut to meet the page limit. I find that individual parts are written quite well, but the paper as a whole appears to be missing some significant pieces.

**Limitations:**

The authors do not discuss the limitations and potential negative societal impact of their work to any noticeable extent. Still, given the topic of research, I do not consider it a defect.

**Strengths And Weaknesses:**

The paper makes a highly interesting theoretical contribution by showing that the "Stochastic Equicontinuity" condition is implied by a stability condition on the estimation algorithm (Lemma 2). This algorithmic stability condition is shown to be weaker than the conditions previously imposed in the literature (Remark 2). The contribution is also interesting from a practical point of view because it provides an explanation of why sample splitting is or is not needed for debiased machine learning.

One criticism about the result itself is that it may be difficult to check the stability conditions for arbitrarily complicated algorithms except in some special cases, e.g., bagged estimators. If this is the case, a practitioner may feel more safe sticking with the sample splitting approach. Also, the paper does not give any experimental results to back up its claims. Since the theoretical results describe the limiting behavior, it would have been helpful to have extensive experimental results to see how the claims hold up at finite sample sizes.

---

> ### Author Response · Authors · 2022-08-02
> **Answers to Reviewer v7GP**
>
> Thank you very much for your input and effort in reviewing the paper. We appreciate your suggestions and comments.
>
> 1.About verifying the conditions of Lemma 2:
>
> Indeed, verifying the conditions might be difficult, but our stability conditions are implied by Algorithmic Stability and $L^{2r}$-continuity (please see Corollary 4). Algorithmic Stability is a classical condition in learning theory literature [1, 2] and holds true for various regularized empirical risk minimization algorithms [3] and stochastic gradient descent algorithms [5]. $L^{2r}$-continuity and the mean-squared-continuity condition in Lemma 2 are regular smoothness conditions and in Appendix Section F we demonstrated how to verify them for causal inference examples. Moreover, we circumvent the need of Donsker property, which is a very strong and impractical assumption commonly assumed in classical work, while requiring a much slower consistency rate than previous work [4]. We also provide a counterexample to show that our stability conditions are tight (please see the next next bullet point). Once we establish stochastic equicontinuity, all the other conditions of Theorem 1 are standard regularity conditions in debiased machine learning literature.
>
>
> 2. Answers to the questions raised:
>
> - There is no real connection between stability and consistency. An estimator can be stable but not consistent (it can converge to another quantity for example). Consistency at a given rate implies stability at the same rate. But note that for Theorem 1 to hold we need faster stability rates than the ones given by the consistency (see lemma 3)
>
> - Thank you for your suggestion to explore experimentally the finite sample effect of our stability condition. This is an interesting idea. We will explore this for the camera-ready version.
>
> - When our conditions are not met the estimators will in general not be asymptotically normal (see lemma 3). We also believe that the bootstrap method (as well as other classical methods) will not be in general consistent. Therefore we sadly do not have recommendations in this context.
>
> - The second to last equality in the unnumbered display equation following Line 183 exploits the tower law.
>
> - The second to last equality in the unnumbered display equation following Line 186 uses the fact that the expression $a(Z_l,g_0)-A(g_0)$ is a function of $Z_l$ only and is thus independent of the data points in $Z^{(-l)}$.
>
> - The last equality in the unnumbered display equation following Line 198 follows because the distribution of $a(Z,\hat g^{(-l_1)})-a(Z,\hat g^{(-l_1,-l_2)})$ is the same as the distribution of $a(Z,\hat g)-a(Z,\hat g^{(-l_2)})$, noting that $Z_{l_2}'$ has the same distribution as $Z_{l_2}$ and they are both independent from all the other data points.
>
> We will include more detailed explanations for these equalities in the camera-ready version.
>
> 3. About the significance of Lemma 3:
>
> In our lemma 2, we showed that $o(\frac{1}{\sqrt{n}})$ stability rate would imply stochastic equicontinuity. However, in Lemma 3, we demonstrated that exactly $O(\frac{1}{\sqrt{n}})$ stability rate, which is just weaker than our stability condition, would not be sufficient for stochastic equicontinuity. This shows that our stability rate is tight and thus necessary when other conditions are kept the same.
>
> 4. About the experimental results:
>
> Actually we have included experimental results in Appendix Section A in our supplementary material. Our experimental results demonstrate the validity of our results (the estimator is approximately Gaussian) and clearly show the advantage of not doing data splitting (even for finite sample size).
>
> 5. About the presentation:
>
> Thanks for pointing this out. We will change our presentation to make it more clear and smooth.
>
> References:
>
> [1] Morgane Austern and Wenda Zhou. Asymptotics of cross-validation. 2020.
>
> [2] Pierre Bayle, Alexandre Bayle, Lucas Janson, and Lester Mackey. Cross-validation confidence intervals for test error. 2020
>
> [3] Olivier Bousquet and Andr{\'e} Elisseeff. Stability and generalization. 2002.
>
> [4] Victor Chernozhukov, Whitney Newey, Rahul Singh, and Vasilis Syrgkanis. Adversarial estimation of riesz
> representers. 2020.
>
> [5] Moritz Hardt, Ben Recht, and Yoram Singer. Train faster, generalize better: Stability of stochastic gradient descent. 2016.

---

### Official Review · Reviewer_Becs · 2022-07-12

**Rating:** 3
**Confidence:** 3
**Soundness:** 2 fair
**Presentation:** 1 poor
**Contribution:** 2 fair

**Summary:**

This paper deals with the following problem:
Suppose we are interested in solving the generalized moment problem

$M := \mathbb{E}_Z m(Z,\theta, \hat{g}) = 0$, in $\theta$.

Here $Z$ is a random variable, $m$ the moment function, $\theta$ a parameter, and $\hat{g}$ an estimator, that is learned from data.
The goal is to find $\hat{\theta}$ that makes the above expectation $M$ small.



Such a situation occurs, for instance, in treatment effect estimation settings.  In that case, $Z$ consists of $Z=(Y,T,X)$, where $Y$ is the outcome, $T$ is the treatment variable, and $X$ additional features. The $\hat{g}$ to be learned from data is an estimator of $T$ and $Y$ as functions of $X$.

Typically, to obtain bounds on the estimation of $\hat{\theta}$, one splits the sample into two parts. Then the first part is used to estimate $\hat{g}$ and the second part is used to estimate the empirical minimizer $\hat{\theta}$ of $M$, based on $\hat{g}$.

In this paper it is shown that if $m$ is a _linear_ function of $\theta$ (e.x.  $Y = \theta T + \ldots$ in example above), and the classifier is chosen by an algorithm that has the algorithmic stability property,  then one can use the full same sample to execute both estimations above, and still have the unbiasedness and asymptotic normality property for the estimator of $\theta$.

It is also shown that bagging type estimators have algorithmic stability under some conditions on the base estimator. As an example of this,
the authors show that bagging ensemble of 1-nearest neighbor classifiers has algorithmic stability and thus can be used as $\hat{g}$ in the main result.



**Questions:**

Please see the discussion above.

**Limitations:**

Limitations and potential negative societal impact were not discussed.

**Strengths And Weaknesses:**

There are three serious issues with this paper.  **(1)** The first issue is that this paper is not well written. In terms of language and focus of presentation I believe it is not suitable for general ML audience in its current form. Moreover, in terms if technical presentation, it is also very hard to read. Definitions and references are hard to find around the paper.
**(2)** The second issue is that the connection between bagging and stability is well known. It is not completely clear why the authors prove these results, and claim them as contributions.  **(3)** The scope of the results is limited. Only linear moment functions are considered, and the results are about just asymptotic estimates rather than finite sample bounds.


To discuss these points in more detail:

(1) This paper will be very hard to read to people who are not familiar with the generalized method of moments, treatment effect estimation and instrumental variables. I believe most NeurIPS participants are not familiar with these notions, although they are very familiar with the underlying mathematics. Therefore I would suggest to the authors to completely rewrite the paper, carefully introduce the problems  and their importance, before discussing the paper's contribution.

(2) Connections between bagging and stability are natural and known. See for instance [a] below. Even if the specific version of the results
in this paper is new, it should be discussed in the general context.  How do the results presented in the paper differ from standard results?
In addition, while 1-NN classifiers are rarely used in ML communities. Thus I would suggest providing more realistic classifier examples. Would regularized kernel regression satisfy the stability assumptions used in the paper?



[a]  Elisseeff, A., Evgeniou, T., Pontil, M., & Kaelbing, L. P. (2005). Stability of Randomized Learning Algorithms. Journal of Machine Learning Research, 6(1).

---

> ### Author Response · Authors · 2022-08-02
> **Answers to Reviewer Becs**
>
> Thank you very much for your input and effort in reviewing the paper. We appreciate your suggestions and comments.
>
> 1. About the presentation style:
>
> Thanks for pointing it out. We will include more background to make the language of our camera-ready version more comprehensible to the general audience.
>
> 2. About the connection between bagging and stability:
>
> Yes the connection is well-established, but it is in fact not the main contribution of our paper. Our main contribution is Lemma 2, which states that under regularity condition, a stability rate of $o(n^{-1/2})$ is sufficient to guarantee stochastic equicontinuity, which will (along with the other conditions of Theorem 1) in turn guarantee asymptotic normality and root-n consistency of the estimation $\hat\theta$, even if the estimation has been done without data splitting. This is interesting because in practice, data splitting greatly reduces data efficiency. The connection between bagging and stability was mentioned only as a concrete example of where the stability conditions are satisfied, to demonstrate applicability of our Lemma 2. We also included the stability proof for bagging for completeness and just in case a reader is not familiar with the stability literature or in case a reader thinks that our stability conditions are not of the same form as in the literature due to discrepancies in notation. In general, our stability conditions also hold for many estimators other than bagging estimators. For example, it is satisfied by a variety of regularized empirical risk minimization algorithms [1] (notably an RKHS will satisfy our conditions as long as the regularizing coefficient doesn't decrease too fast $\lambda>>n^{-1/2}$)  and stochastic gradient descent algorithms [4]. We will mention these examples in our camera-ready version. Thanks for the advice.  However note that for Theorem 1 to hold we need also need to establish consistency at rate of $o(n^{-1/4})$.  Finally note that our stability condition is tight (see lemma 3), which we demonstrate with a counter-example.
>
> 3. About the linear moment assumption:
>
> Thanks for pointing this out. Yes, we agree that this is a restriction, and to alleviate this we have included an extension of our results to the nonlinear moment case (please see the new Appendix Section H for the revised version). We omitted working out such an extension from the original submission for several reasons: 1) it introduces more technical notation and more technical regularity conditions, 2) it requires slightly stronger uniform in $\theta$ stability conditions for consistency, 3) it requires a separate technical consistency lemma, as is typical for nonlinear moments, 4) the asymptotic normality (root-n rate) result, as is standard for nonlinear moments, only starts to kick in once the estimate $\hat\theta$ is sufficiently close to $\theta_0$, which always happens after some sufficiently large sample size, but that sample size could be large and not well controlled. This is a typical behavior of nonlinear moment proofs, which makes them more brittle in finite samples. Moreover, we would also like to point out that in practice, most causal inference problems use linear moments [2, 3, 5]; and that many recent papers, including [6] and [7], impose even stronger restrictions on the moments functions. Most widely used in practice instantiations of the debiased machine learning (DML) framework involve linear moments. For instance, both the dml package in R and the econml package in python (used in industry) only implement linear moment cases of the DML framework.
>
> 4. About the asymptotic nature of the results:
>
> Yes the results we presented are asymptotic. But we believe that our stability conditions could also be used to establish finite sample guarantees on the form of Berry-Esseen bounds. Moreover our experiments (see section A of the appendix) show that in practice our result also hold in finite sample: the estimator is almost normal even without doing data splitting.
>
> References:
>
> [1] Olivier Bousquet and Andr{\'e} Elisseeff. Stability and generalization. 2002.
>
> [2] Victor Chernozhukov, Denis Chetverikov, Mert Demirer, Esther Duflo, Christian Hansen, Whitney Newey, and
> James Robins. Double/debiased machine learning for treatment and structural parameters, 2018.
>
> [3] Victor Chernozhukov, W Newey, James Robins, and Rahul Singh. Double/de-biased machine learning of global
> and local parameters using regularized riesz representers. 2019.
>
> [4] Moritz Hardt, Ben Recht, and Yoram Singer. Train faster, generalize better: Stability of stochastic gradient descent. 2016.
>
> [5] Peter M Robinson. Root-n-consistent semiparametric regression. 1988.
>
> [6] Victor Chernozhukov, Whitney K. Newey, and Rahul Singh. A simple and general debiased machine learning theorem with finite sample guarantees. 2021.
>
> [7] Victor Chernozhukov, Whitney K. Newey, and Rahul Singh. Automatic debiased machine learning of causal and structural effects. 2022.

---

### Meta-Review · Area_Chair_ffw4 · 2022-08-22

**Recommendation:** Accept
**Confidence:** Certain

**Metareview:**

The paper provides an analysis that avoids sample-splitting while estimating parameters with objective functions involving other estimated parameters. The results are interesting and the analysis is insightful.

One of the reviewers (Reviewer Becs) raised some concerns about style of the writing, high-level questions on (bagging and stability) and linear moment assumptions. In my reading of the paper, I found no significant issues about the style (in fact the paper is well-written from a statistical point-of-view). The authors addressed the other two issues in the rebuttal. Hence, the score of reviewer Becs was down-weighted and the paper is recommend for acceptance at Neurips.

**Award:**

No

---

### Decision · Program_Chairs · 2022-09-14

Accept